# Robust Power System State Estimation Method Based on Generalized M-Estimation of Optimized Parameters Based on Sampling

**Yu Shi** [1]**, Yueting Hou** [2]**, Yue Yu** [2]**, Zhaoyang Jin** [2,*] **and Mohamed A. Mohamed** [3]

1    Department of Science, Shandong Jiaotong University, Jinan 250353, China
2    Department of Electrical Engineering, Shandong University, Jinan 250100, China
3    Department of Electrical Engineering, Faculty of Engineering, Minia University, Minia 61519, Egypt
*    Correspondence: zhaoyang.jin@sdu.edu.cn; Tel.: +86-178-6299-7751

**Abstract:** Robustness is an important performance index of power system state estimation, which is defined as the estimator's capability to resist the interference. However, improving the robustness of state estimation often reduces the estimation accuracy. To solve this problem, this paper proposes a power system state estimation method for generalized M-estimation of optimized parameters based on sampling. Compared with the traditional robust state estimator, the generalized M-estimator based on projection statistics improves the robustness of state estimation, and the proposed optimized parameter determination method improves the overall accuracy of state estimation by appropriately adjusting its robustness. Considering different degrees of non-Gaussian distributed measurement noises and bad data, the estimation accuracy the proposed method is demonstrated to be up to 23% higher than the traditional generalized M-estimator through MATLAB simulations in IEEE 14, 118 bus test systems, and Polish 2736 bus system.

**Keywords:** Gaussian distribution; M-estimator; power system state estimation; precision; robustness; weighted least square method

## 1. Introduction

Modern power systems need to grasp the real-time operation of the power system comprehensively and quickly. It can more accurately analyze and predict the system's operation trends to ensure the power system's economy and safety. Power system state estimation is an effective means to detect the real-time operation data of the power system which was first proposed by the Schweppe in 1970 [1]. Abundant research has been conducted on power system state estimation to optimize the real-time redundant measurement data of the power system, to realize the real-time and reliable monitoring of the power system, and to ensure the safe and reliable operation of the system [2–8].

The weighted least squares (WLS) method has been widely adopted since 1970, with the advantages of good convergence performance and accurate estimation results. Its disadvantage is that it is not sufficiently robust to measured noise that follows the Gaussian distribution without bad data interference. In the real power network, not all measurement noise can be represented strictly by a Gaussian distribution, and occasionally significant errors occur due to various known telemetry noises or faults. There may also be other types of outliers that strongly affect the estimated state but may or may not carry bad data. All of these factors can potentially affect the estimator's estimation accuracy. A series of studies have been conducted to improve the robustness of state estimation, which is summarized in Table 1:

**Table 1.** Reference summary.

| Authors | Origin | Purpose | Advantages and Disadvantages |
|---|---|---|---|
| L. Mili, M. G. Cheniae, N. S. Vichare and P. J. Rousseeuw | USA | To describe a fast and robust method for identifying the leverage points [9]. | The method is very fast and compatible with real-time applications, but it does not apply to all forms of lever points. |
| J. Zhao and L. Mili | USA | To develop a robust dynamic state estimator of a cyber-physical system [10]. | The H-infinity filter is able to handle large system uncertainties as well as suppress outliers, but the estimation efficiency of this method is low. |
| M. B. Djukanovic, M. H. Khammash and V. Vittal | USA | To present a framework for robust stability assessment in multimachine power systems [11]. | The proposed method significantly reduces computational complexity and at the same time preserves the accuracy in predicting stability robustness. |
| Z. Lyu, H. Wei, X. Bai, D. Xie, L. Zhang and P. Li | CHN | To propose an norm estimator [12]. | The proposed estimator has high computational efficiency and robustness. |
| E. Kyriakides, S. Suryanarayanan and G. T. Heydt | USA | To demonstrate the Huber function technique in a power engineering application [13]. | This technique reduces large residuals but not accuracy. |
| M. Göl and A. Abur | TR | To develop a PMU placement strategy [14]. | This method can improve the stability and accuracy of estimation. |
| M. Netto, J. Zhao and L. Mili | USA | To develop a robust extended Kalman filter [15]. | The robust extended Kalman filter exhibits good tracking capabilities under Gaussian process and observation noise while suppressing observation outliers, even in position of leverage. However, it presents poor performance under non-Gaussian noise. |
| I. Akingeneye, J. Wu and J. Yang | USA | To develop PMU placement algorithms to improve the power grid state estimation [16]. | The performance of the low complexity algorithms approach that of the exhaustive search algorithm, but with a much lower complexity. |
| G. Wang, G. B. Giannakis and J. Chen | USA | To put forward a novel LAV estimator leveraging recent algorithmic advances in composite optimization [17]. | The algorithm efficiently deals with the non-convexity and non-smoothness of LAV-based PSSE, but it relies on solving a sequence of convex quadratic subproblems. |
| M. Huang, Z. Wei, G. Sun and H. Zang | CHN | To propose a hybrid SE for distribution systems [18]. | The estimator method provides more reliable estimation results with a limited number of SCADA measurements, while biased estimated results can exist if some buses are far away from the measuring points. |
| C. H. Ho, H. C. Wu, S. C. Chan and Y. Hou | CHN | To present a robust statistical approach [19]. | The proposed approach outperforms conventional approaches using the ADMM with L1 outlier detection in state estimation accuracy and convergence speed. |
| J. Zhao, M. Netto and L. Mili | USA | To develops a robust iterated extended Kalman filter based on the generalized maximum likelihood approach [20]. | GM-IEKF can achieve both robustness and statistical efficiency, but its vulnerability to system parameter and topology errors. |

The problem of existing robust state estimators is that robustness of the state estimator is usually achieved at the expense of estimation accuracy. The reason is that the existing robust state estimators use fixed parameter settings, which cannot adjust the state estimator's robustness and estimation accuracy in different scenarios. In [21], it is indicated that the measurement noise in PMU is likely to be non-Gaussian, which leads to the increased probability of outliers (measurements that significantly departs from their true values). Moreover, the types, parameters, and proportions of the non-Gaussian measurement distribution are different in different systems. Therefore, existing robust state estimators will be less robust when bad data and outliers are more frequent and will have insufficient accuracy when bad data and outliers are less frequent. To overcome this problem, this paper first evaluates the impact of bad data and non-Gaussian measurement noises on existing generalized

M-estimation parameters based on extensive Monte Carlo simulations. Finally, this paper proposes a generalized M-estimator of optimized parameters based on sampling which can adaptively select the appropriate parameters according to the probability distribution of the system measurement noise, accurately identify the outliers, and significantly reduce the effects of non-Gaussian measurement noise and bad data.

The contributions of this paper are summarized as follows:

- Demonstrate that higher robustness does not necessarily improve the estimation accuracy of the state estimator, and the best accuracy can be achieved if the robustness is tuned at an appropriate level;
- Propose a new robust power system state estimation method that can adaptively tune its robustness according to different levels of non-Gaussian distributed measurement noise and bad data.

The paper is structured as follows. Section 2 introduces the existing robust state estimation methods. Section 3 describes the drawbacks of existing robust state estimation methods. Section 4 presents the proposed generalized M state estimator of optimal parameters based on sampling. Section 5 presents and discusses the simulation results. Finally, Section 6 concludes this paper and identifies opportunities for future developments.

## 2. Existing Robust State Estimators for Power System Estimation

This section presents two widely used robust state estimators for power system state estimation, M-estimator (including Huber estimator and LAV estimator) and generalized M-estimator, which is improved based on the M-estimator.

### 2.1. M-Estimation of the Static State Estimation Method

The concept of M-estimation was first used by Huber for robust estimation of distribution centers and subsequently generalized to regression [22]. In general, an M-estimator is a maximum-likelihood estimator. It minimizes an objective function that is expressed as a function $\rho(\mathbf{r})$ for measuring the residue, according to the constraints given by the measurement equation:

Objective function:

$$\sum_{i=1}^{m} \rho(r_i) \tag{1}$$

Constraint condition:

$$\mathbf{z} = \mathrm{h}(\mathbf{x}) + \mathbf{r} \tag{2}$$

where $\rho(r_i)$ is a selected function that can measure the residual $r_i$, z is the measurement vector, $\mathbf{x}$ is a state vector and, h($\mathbf{x}$) is a measurement function.

The Huber estimator objective function is expressed as:

$$J(\mathbf{x}) = \sum_{i=1}^{m} \omega_i^2 \rho(r_i) \tag{3}$$

where the Huber function, $\rho(r_i)$, can be defined as:

$$\rho(r_i) = \begin{cases} r_i^2/2 & \text{for } |r_i| \leq \beta \\ \beta|r_i| - \beta^2/2 & \text{for } |r_i| > \beta \end{cases}. \tag{4}$$

The first partial derivative $\psi(r_i)$ of $r_i$ is expressed as:

$$\psi(r_i) = \begin{cases} r_i, & |r_i| < \beta \\ \beta * sign(r_i), & else \end{cases}. \tag{5}$$

The least absolute value (LAV) function can be defined as:

$$\rho(r_i) = |r_i| \tag{6}$$

where $r_i$ is the normalized residual, the parameter $\beta$ is a fixed value, generally set to 1.5. In fact, if $\beta$ is set to infinite, the Huber estimator is equal to WLS estimator parameters, which has the highest accuracy. If $\beta$ is set to 0, the Huber estimator turns into a LAV estimator, which has the highest robustness. Therefore, the Huber estimator can be regarded as an estimator which strikes a balance between the advantages of the WLS estimator and the LAV estimator.

The loss functions of the Huber and the LAV estimator are shown in Figure 1:

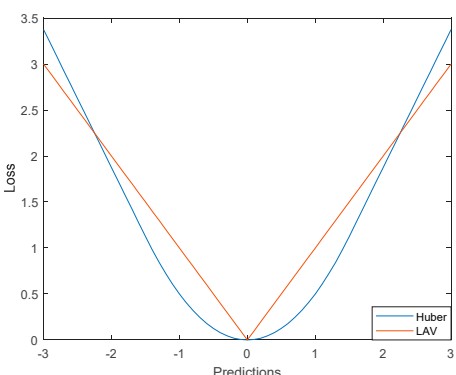

**Figure 1.** Loss functions of the Huber and LAV estimators.

### 2.2. Generalized M-Estimator

The external interference of the system causes outliers. There are multiple types of external interference. For example, there can be errors in sampling, such as recorded bias, calculation errors, etc. It can also be caused by various accidental and abnormal factors. The presence of outliers affects the accuracy of state estimation. Although the Huber estimator and the LAV estimator have better robustness compared to the WLS estimator, it is still obviously affected by outliers. Therefore, early robust estimators suppress the effect of outliers by reducing their weights according to their normalized residuals.

The normalized residual is based on Mahalanobis distance, which is the weighted average distance between each point. The disadvantage is that the average point is significantly affected by outliers, leading to an inaccurate judgment of outliers. In contrast, the median is hardly affected by the outliers. However, the variance of each point to the median cannot be directly defined. It is more reasonable to use the projection statistics method. The idea of the projection statistics method is to study those coming from the coordinated median **M**, and through each data point direction v. Formally, we have v = $l_i - $ **M**, where:

$$\mathbf{M} = \left[ \underset{j}{median}\mathbf{l}_{j1}, \cdots, \underset{j}{median}\mathbf{l}_m \right]^T \tag{7}$$

where $j$ is obtained from 1 to $m$. The resulting distance calculated with (7) will be referred to as the projection statistics, or the $PS_i$ for short. Then, the $i$-th calculated projection statistic, $PS_i$, is compared to a given threshold. The labeled outliers are then degraded using the following weight function: $\omega_i^2 = \min\left(1, d^2/PS_i^2\right)$, where $d = 1.5$ is set to produce good statistical efficiency.

The generalized M-estimator combines outlier detection and the Huber loss function, whose objective function is expressed as:

$$J(\mathbf{x}) = \sum_{i=1}^{m} \omega_i^2 \rho(r_s). \tag{8}$$

To solve the objective function, we calculated the first derivative about **x**:

$$\frac{\partial J(x)}{\partial x} = \sum_{i=1}^{m} -\frac{\omega_i a_i}{s} \psi(r_{si}) = 0 \tag{9}$$

where $s = 1.4826 \, median_i \, |r_i|$.

It is clear that this is a set of equations which can be solved by iterated re-weighted least square (IRLS) algorithm [23].

## 3. Disadvantages of Existing Robust State Estimators for Power System State Estimation

Existing robust estimators mainly consider the suppression performance on bad data, while the assessment of their outlier suppression performance caused by non-Gaussian measurement noise is insufficient. This might result in a decreased estimation accuracy of the state estimator in the presence of non-Gaussian measurement noise. The two most common types of non-Gaussian measurement noise are bimodal Gaussian distribution and Laplace distribution, which are introduced in detail in Section 3.1. Section 3.2 discusses the impact of non-Gaussian measurement noise on the performance of existing robust estimators. Section 3.3 presents the purpose of this study.

### 3.1. Non-Gaussian Distributed Measurement Noises

3.1.1. Bimodal Gaussian Distribution

In practice, the errors of voltage and current measurements might follow bimodal Gaussian mixture (BGM) distribution [24]. The probability density function can be obtained by the superposition of two Gaussian probability density functions:

$$PDF_{BGM} = \prod_{i=1}^{2} \omega_i f_{N(\hat{\mu}_i, \hat{\sigma}_i^2)}(y) \tag{10}$$

where $f_N(y)$ represents the Gaussian probability density function, $N(\mu, \sigma^2)$ represents the normal distribution, and $\omega$ represents the weights corresponding to the combination of Gaussian components. The symbol "ˆ" represents the estimated quantity, and the subscript $i$ represents the $i$-th Gaussian component combination. The weight of each Gaussian component combination is the product of all the Gaussian component weights involved in that combination, satisfying:

$$\omega_1 + \omega_2 = 1. \tag{11}$$

For the bimodal mixed Gaussian distribution in the method, the mean and variance are as follows:

$$\mu_{12} = \omega_1 \mu_1 + \omega_2 \mu_2 \tag{12}$$

$$\sigma_{12}^2 = \omega_1 \sigma_1^2 + \omega_2 \sigma_2^2 + \omega_1 \omega_2 (\mu_1 - \mu_2)^2 \tag{13}$$

where $\omega$ represents the weight, $\mu$ represents the mean value, $\sigma$ represents the variance, subscripts 1 and 2 indicate the components 1 and 2 of the BGM, respectively, and subscript 12 indicates the resulting BGM distribution. In the proposed method, $\mu_{12}$ is set to 0, which results in

$$\frac{\mu_1}{\mu_2} = -\frac{\omega_2}{\omega_1} = k. \tag{14}$$

Combine (12) and assume $\sigma_1 = \sigma_2 = \sigma$, we have,

$$\mu_1 = -k\omega_2 \sigma \tag{15}$$

$$\mu_2 = k\omega_1 \sigma. \tag{16}$$

Therefore, the distribution of the BGM measurement noise can be changed by adjusting the parameter, $k$.

### 3.1.2. Laplace Distribution

The Laplace distribution is also called bi-exponential distribution because it can be seen as the combination of two exponential distributions at different positions. The probability density function of the Laplace distribution is:

$$f(x|\mu, b) = \frac{1}{2b} \exp(-\frac{|x - \mu|}{b}) = \frac{1}{2b} \begin{cases} \exp(-\frac{x-\mu}{b}), if \ x \geq \mu \\ \exp(-\frac{\mu-x}{b}), if \ x < \mu \end{cases} \tag{17}$$

where $\mu$ is the position parameter, $b > 0$ is the scaling parameter. If $\mu = 0$, then the positive half happens to be an exponential distribution of scale $1/2$. The difference between Laplace distribution and Gaussian distribution is that the Gaussian distribution represents the square of the difference relative to the mean, while the Laplace distribution is represented by the absolute value relative to the difference [25]. Thus, the tail Laplace distribution is much flatter than that of the Gaussian distribution.

### 3.2. Effect of Non-Gaussian Measurement Noises on the Performance of Existing Robust State Estimators

State estimation aims to determine the most likely states in the system based on the measurements. One way to achieve this is through maximum likelihood estimation (MLE). Assuming that all measurement noises have known probability distributions, all measurements' joint probability density function can be written with these unknown parameters, called the likelihood function, which will peak when the unknown parameter is chosen closest to its actual value. When the measurement noise follows the Gaussian distribution, the deduced optimal solution method is the WLS [26].

As shown in Figure 2, WLS is no longer the optimal solution when the measurement noise does not follow the Gaussian distribution. This is because, compared to the Gaussian distribution, the tail convergence rate of the bimodal Gaussian distribution and the Laplace distribution are faster than that of the Gaussian distribution tail convergence rate (so they are also known as a heavy tail distributions). Therefore, they have higher outlier probability than the Gaussian distribution, i.e., their absolute value in the white part is greater than that of the Gaussian distribution. If WLS is still used in the existence of non-Gaussian distributed noises, outliers will be assigned with large weights, resulting in reduced estimation accuracy.

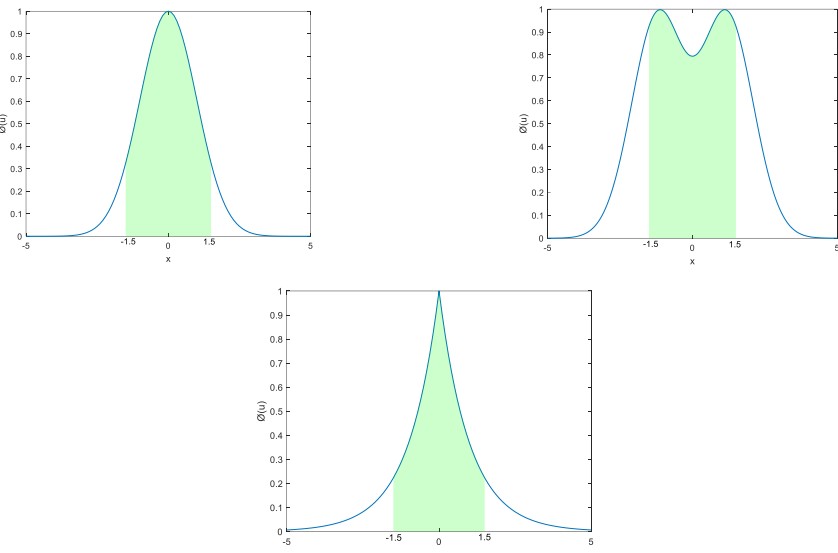

**Figure 2.** Probability density functions of Gaussian, bimodal Gaussian, and Laplacian distributions.

In discussing the effect of non-Gaussian measurement noise on the performance of existing robust state estimators, this subsection only discusses M-estimators since the

generalized M-estimator is still developed based on the M-estimators. As shown in (5), the value of the target function of LAV is proportional to the absolute value of the deviation value, so when the deviation value is large when the outlier appears, the weight is much less than the WLS, so it has high robustness for the outliers. Although there may be significant proportion of non-Gaussian measurement noises, the Gaussian measurement noise still constitutes the main part for real systems. Therefore, the LAV estimator might not achieve the highest estimation accuracy.

The Huber estimator combines the advantages of both WLS and LAV according to (4). The size of the parameter $\beta$ determines Huber's emphasis on WLS and LAV: when $\beta \rightarrow \infty$, Huber estimator is equivalent to WLS; when $\beta = 0$, it is equal to LAV. The parameter $\beta$ is generally set to be the fixed value of 1.5. Therefore, the existing Huber estimators and generalized M-estimators based on fixed $\beta$ values may have insufficient robustness or accuracy in the existence of the non-Gaussian distribution measurement noise with different proportions and parameters. Therefore, it is urgent to propose a generalized M-estimator (or Huber estimator) that optimizes the value according to the measurement noises.

### 3.3. Research Purpose of Generalized M State Estimation of Optimization Parameters Proposed

As shown in the above sub-sections, non-Gaussian distributed measurement noises are more likely to cause outliers than the Gaussian distributed measurement noises. Therefore, different levels of robustness of the estimator are required to ensure the best estimation accuracy under different types and proportions of non-Gaussian distributed measurement noises. Traditional generalized M-estimation based on Huber estimation with fixed parameter $\beta$ can ensure high robustness against bad data and high estimation accuracy under Gaussian distributed noise. Its estimation accuracy is decreased if the measurement noises are non-Gaussian distributed. So, the purpose of this study is to propose a method that can adaptively change the value of $\beta$ so that the generalized M-estimator can have high estimation accuracy under different non-Gaussian distributed measurement noises while ensuring good robustness against bad data.

## 4. Generalized M State Estimator of Optimized Parameters Based on Sampling

To overcome the drawback of the traditional generalized M-estimator, a novel generalized M state estimator of optimized parameters based on sampling is proposed in this section, where the random sampling method is described in Section 4.1, and the algorithm process of sampling-based optimization parameter is introduced in Section 4.2.

### 4.1. Optimized Parameter Selection Method Based on the Random Sampling Method

In Section 3.2, it is explained that non-Gaussian noise has a higher outlier probability than Gaussian noise. Hence, the value of $\beta$ needs to be reduced to increase the estimator's robustness. Because the optimal value is affected by many uncertain factors, such as the configuration of measurement noise, the probability distribution of measurement noise, and the probability of bad data occurrence, it is almost impossible to obtain the exact optimal value of $\beta$. Therefore, this paper proposes the sagging optimization selection method to find a value of β close to its optimal value according to the probability ratio of the outliers:

$$\beta = 1.5 - a \frac{\sum\limits_{i=1}^{N} P_{Truthi}(x \geq 1.5)}{\sum\limits_{i=1}^{N} P_{Gaussi}(x \geq 1.5)}. \tag{18}$$

Define $n_1 = \sum\limits_{i=1}^{N} P_{Truthi}(x \geq 1.5)$, which is the number of times when the actual standardization error ((measurement value-mean)/standard deviation) of all the measurements in the system is greater than 1.5, is the quantity to be determined. Define

$n_2 = \sum\limits_{i=1}^{N} P_{Gaussi}(x \geq 1.5)$, which is the number of times when all the measurements obey the Gaussian distribution, is the known quantity. The variable $a$ is the droop coefficient and the quantity to be determined. This paper takes the following steps to determine $n_1$ and $a$:

Step 1: for a system with m measurements, randomly select $N$ measurements;

Step 2: For the $N$ measurements, record L groups of data from their corresponding devices (L is large enough), and record the number of normalized errors greater than 1.5, $n_1$.

Step 3: Calculate the droop coefficient $a$ by (19):

$$a = (1.5 - \beta_{\min})\frac{n_2}{n_{1,\max}} \tag{19}$$

where $\beta_{\min}$ is the set minimum value. In order to ensure the convergence ability of the estimator, this paper takes $\beta_{\min} = 0.1$, which is a conservative value. $n_1,\max$ is the number of all measurements whose normalized errors are greater than 1.5 that are calculated from the probability density function at the highest non-Gaussian degree.

If most of the non-Gaussian distribution measurements in the system obey a bimodal Gaussian distribution, then the formula for the optimization values can be approximately reduced to:

$$\beta = 1.5 - Akh \tag{20}$$

where $A$ is the droop coefficient to be sought, and $k$ is the average bimodal Gaussian error coefficient, which is defined in Equation (14), as the non-Gaussian measurement ratio. This paper takes the following methods to determine $A$, $k$, and $h$, as shown in Figure 3.

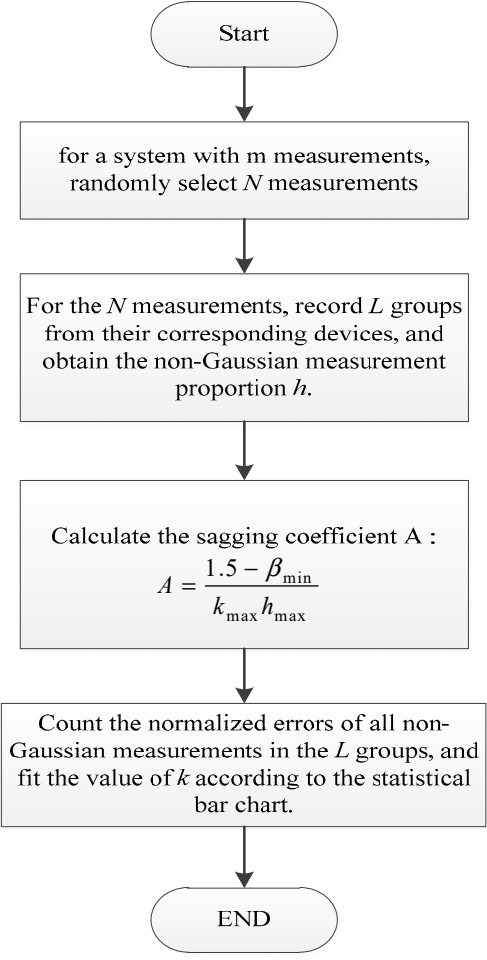

**Figure 3.** The process defined by $A$, $k$, and $h$.

*4.2. The Proposed Generalized M State Estimation Algorithm of Optimized Parameters Based on Sampling*

Based on the optimization parameter selection method proposed in the previous section, this paper proposes an improved generalized M state estimation algorithm. The steps are shown in Figure 4.

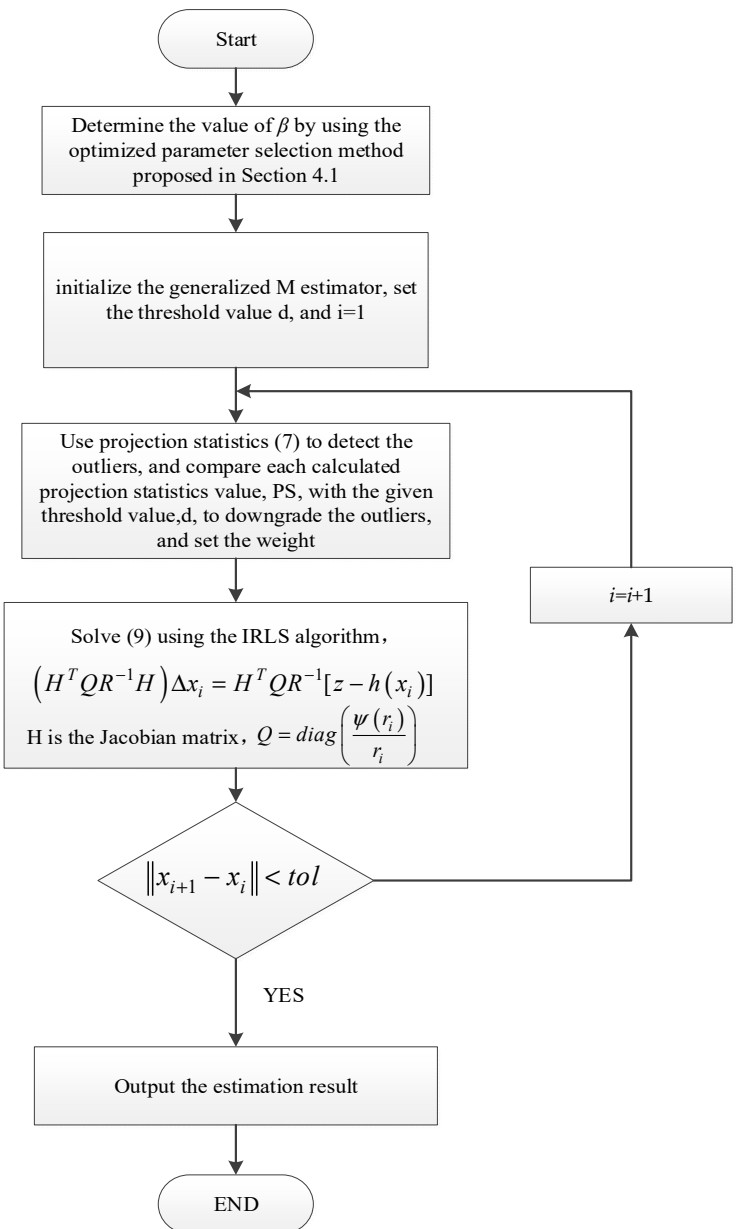

**Figure 4.** The flowchart of generalized M state estimation algorithm for optimization parameters based on sampling.

## 5. Simulated Examples

To verify the effectiveness of the proposed method, the IEEE 14 and 118 bus test systems are selected to simulate the performance generalized M-estimator with different parameters. All simulations were conducted in MATLAB, using an Intel Core i7-9750CPU (@2.6 Hz) 16 GB memory computer. Section 5.1 describes the impact of bad data and non-Gaussian measurement noise on generalized M-estimators with different values of $\beta$ by conducting exhausted Monte Carlo simulations in the IEEE 14 bus test system.

Section 5.2 demonstrates the effectiveness of the proposed generalized M-state estimator of the optimized parameter in a large system, the IEEE 118 bus test system.

### 5.1. Effect of Bad Data and Measurement Noise on the Performance of the Generalized M-Estimator

To ensure the generality of the results, all of the results shown in this section are obtained by averaging the results over the 500 Monte Carlo simulations. To ensure the validity of the generalized M-state estimation method, a set of redundant conventional measurements and PMU measurements were selected in IEEE 14 bus test system as shown in Figure 5.

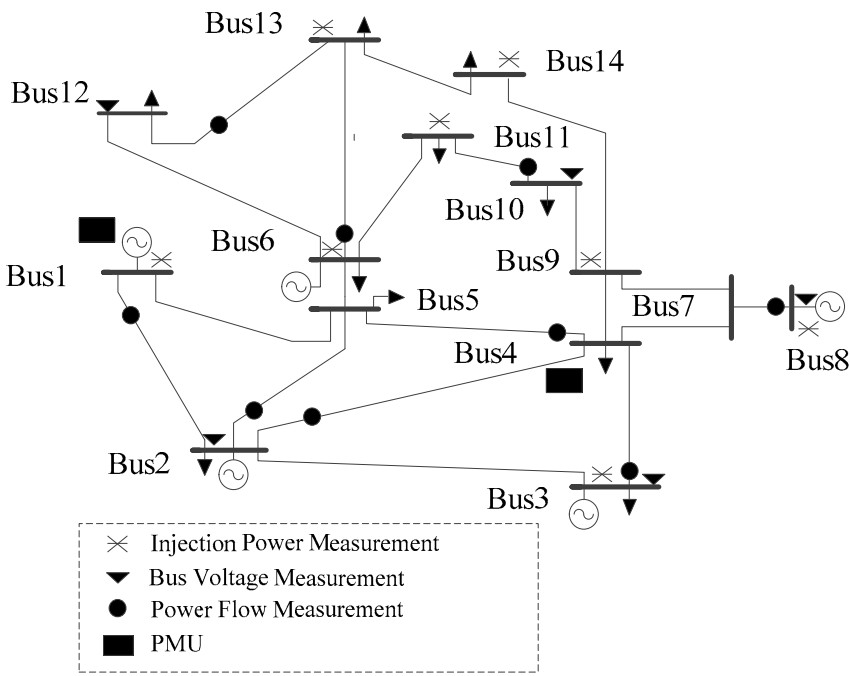

**Figure 5.** Diagram of IEEE 14 test system and measurement configuration.

For Gaussian errors, the standard deviation is set to a percentage of the measurement according to the type of measurement as follows:

SCADA: P, Q measured value (0.02), voltage (0.002)

PMU: Voltage amplitude (0.002), Voltage phase angle (0.01)

The performance of the generalized M-estimator can be adjusted by varying values of $\beta$. In this simulation test, the least value of $\beta$ is $10^{-2}$. With this configuration, the generalized M-estimator can be considered to be an LAV estimator. The largest value of $\beta$ is set to be $10^{4}$. The corresponding estimator can be considered to be a WLS estimator. In the range of $10^{-2}$ to $10^{4}$, $\beta$ increases at the fixed ratio of $10^{1/30}$, all estimators with different values of $\beta$ are tested in the IEEE 14 bus test system. The root mean square error (RMSE) is used as the performance index of the estimators.

Case 1: Existence of bad data with fixed percentage errors. We compared the robustness of the state estimators by introducing fixed percentage errors to the voltage magnitude measurement at bus 1 and obtaining the RMSE of the estimators when the noise probability density function follows the standard Gaussian distribution. The voltage magnitude measurement at bus 1 is changed to 0, 0.2, 0.4, 0.6, 0.8 of its true value (100%, 80%, 60%, 40%, 20% errors, respectively), and tested separately. The simulation results are shown in Figure 6.

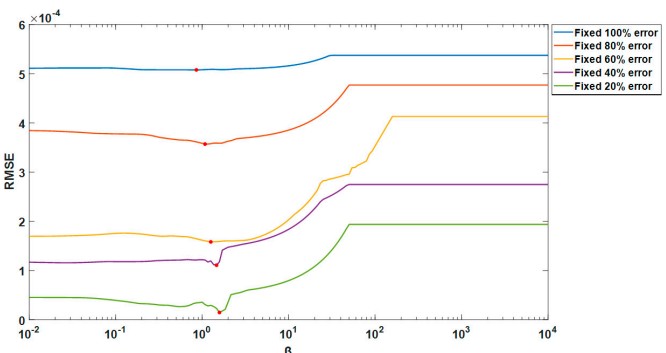

**Figure 6.** The trend of RMSE of each estimator with different β values at fixed error parameter percentages from 0.2 to 1. The red dot is the nadir.

The following conclusions can be obtained:

(1) After introducing fixed errors of different magnitudes, the trend of RMSE of each curve is roughly the same: the general trend is that RMSE increases as a larger $\beta$ is selected. It will not change after a certain value;

(2) There is an optimal value of $\beta$ for each fixed percentage error, and the value decreases with the percentage of the error parameter, as summarized in Table 2.

**Table 2.** Optimal $\beta$ values of each estimator with different fixed error parameters.

| Percentage of Fixed-Error Parameters | 100% | 80% | 60% | 40% | 20% |
|---|---|---|---|---|---|
| The Optimal $\beta$ Value | 0.3415 | 0.5412 | 0.6813 | 0.8577 | 2.3263 |

Case 2: Existence of bad data with random percentage of errors. The robustness of generalized M state estimators with different values of $\beta$ is compared by five individual simulations with three random bad data points in the measurements and observing their performances. The test results are shown in Figure 7.

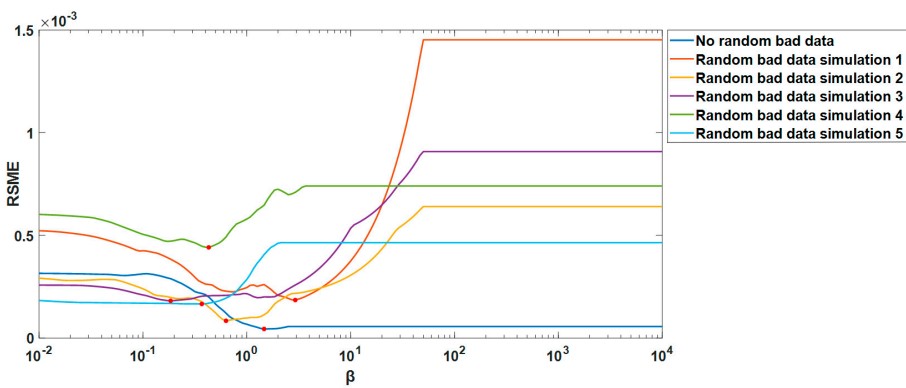

**Figure 7.** The trend of RMSE of each estimator with different $\beta$ values with random error parameter percentages. The red dot is the nadir.

The following conclusions can be obtained:

(1) When there are no bad data in the measurements, the trend of RMSE with the change of $\beta$ is decreasing in general; the value of $\beta$ which achieves the lowest RMSE is 1.5, which is the exact value of $\beta$ for normal generalized M-estimator;

(2) After the introduction of three random errors, the trend of the RMSE of each simulation with the change of $\beta$ is roughly the same: The general trend is increasing. It will not change after a certain value;

(3) There is an optimal $\beta$ whose corresponding estimator achieves the lowest RMSE for each simulation, which varies between $10^{-1}$ and $10^{1}$ with different bad data.

Case 3: Existence of measurements with bimodal Gaussian distribution noise. When the bimodal interval size k (as introduced in Section 3.1) is between 0~5, the trend of the RMSE with estimators with different values of β is shown in Figure 8.

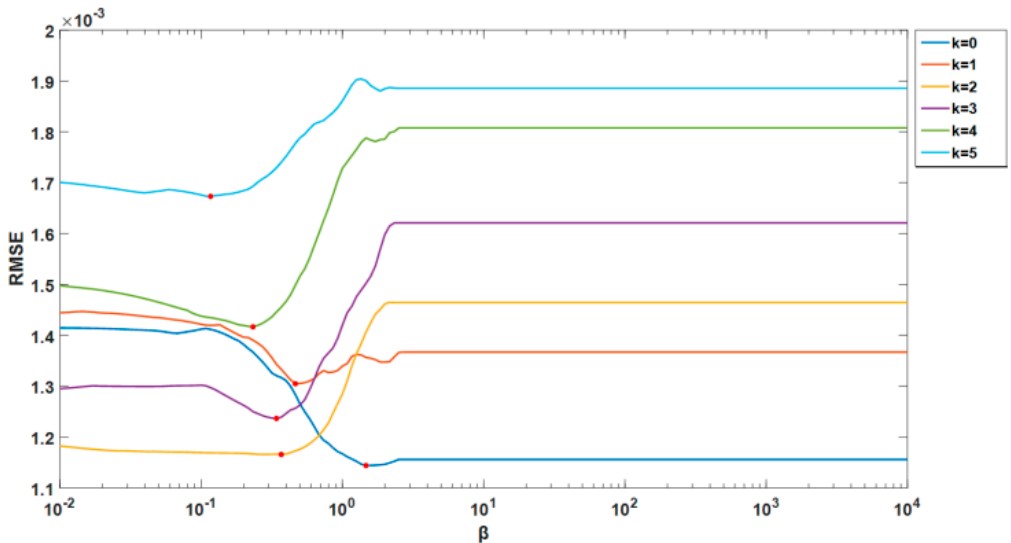

**Figure 8.** The trend of RMSE of each estimator with different $\beta$ values in the presence of bimodal Gaussian distribution noise with bimodal intervals. The red dot is the nadir.

The following conclusions can be obtained:

(1) When $k \leq 1$, RMSE decreases with the increase of $\beta$, and when $k > 1$, RMSE decreases with the increase of $\beta$;

(2) The optimal $\beta$ whose corresponding estimator achieves the lowest RMSE decreases with the increase of $k$;

(3) In the presence of non-Gaussian measurement noise, the value range of the optimal $\beta$ is between $10^{-1}$ and $10^{1}$, which can roughly cover the value range of the optimal β in the presence of bad data;

(4) Since the occurrence of bad data is low and the value range of the optimal $\beta$ in the bad data case can be roughly covered by that in the non-Gaussian measurement noise case, the optimal $\beta$ can be determined by only considering the influence of non-Gaussian measurement noise.

The optimal $\beta$s obtained by simulation in the presence of measurement noises in bimodal Gaussian distribution with different values of $k$ are summarized in Table 3 and plotted in Figure 9. The optimized $\beta$s calculated according to the proposed optimized parameter selection method (20) are also plotted in Figure 9. It can be seen that with the increase of $k$, the calculated optimized $\beta$ decreases, and are equal to the optimal $\beta$ at $k = 0$ and $k \approx 5$. Although the optimized $\beta$ is larger than the optimal $\beta$ for $0 < k < 5$, the decreasing trend guarantees significantly better estimation accuracy compared to the estimator with a fixed $c$ value while $k$ increases.

**Table 3.** Optimal $\beta$ values in the presence of different bi-modal Gaussian distributed measurement noises.

| The Bimodal Interval Size (k) | 0 | 1 | 2 | 3 | 4 | 5 |
|---|---|---|---|---|---|---|
| The Optimal $\beta$ | 1.5000 | 0.4642 | 0.2712 | 0.1849 | 0.1324 | 0.1000 |

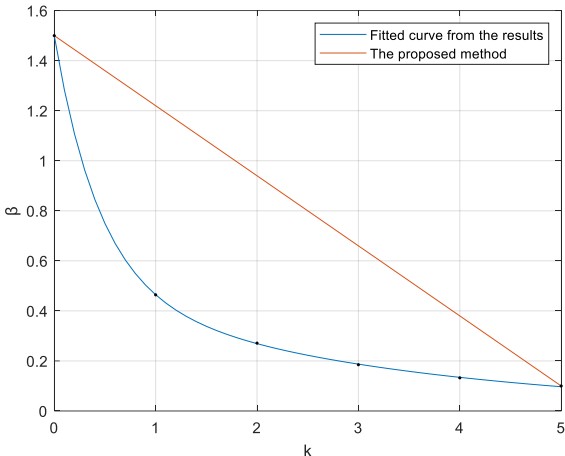

**Figure 9.** A plot of the optimal value and optimized value of $\beta$ against the value of $k$.

### 5.2. Simulation Examples of the IEEE118 Bus Test System

To verify the proposed generalized M-estimation method of optimized parameters based on sampling, simulations are conducted in the IEEE 118 bus test system with the network diagram and measurement configuration shown in Figure 10.

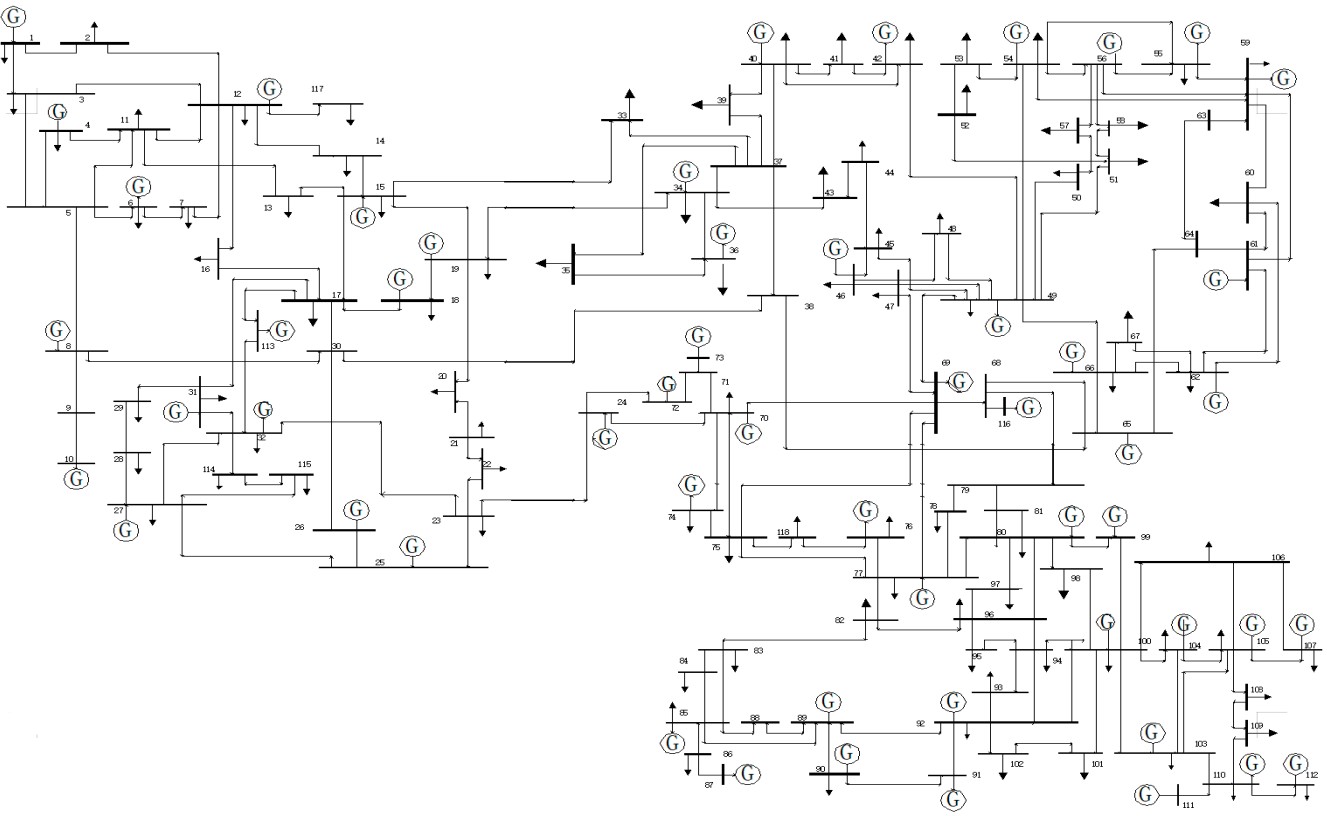

**Figure 10.** Diagram for IEEE 118 test system.

Among the total 186 measurements in the system, 28 of them have bimodal Gaussian distributed noises with the bimodal interval size (k) randomly set between 0~5. According to the proposed optimized parameter selection method, the optimized $\beta$, $\beta$ opd, is determined as follows:

Step 1: Randomly select 80 measurements;

Step 2: For each measurement selected, test the measurement noises and record 10,000 groups of data; 12 non-Gaussian measurement noises are counted according to the statistics, and the proportion of non-Gaussian measurement, *h*, is 15%;

Step 3: Calculate the sagging coefficient A:

$$A = \frac{1.5 - \beta_{\min}}{k_{\max}h_{\max}} = \frac{1.5 - 0.1}{5 \times 0.5} = 0.56$$

where $k_{\max}$ is the maximum bimodal Gaussian error coefficient possible, and $h_{\max}$ is the maximum proportion of the bimodal Gaussian distribution measurement noise possible.

Step 4: Find the average k according to the statistic bar chart of all bimodal Gaussian distributed measurements noises. To test the proposed method in different scenarios, 5 different simulations with average $k \approx 1, 2, 3, 4, 5$ are conducted with the statistical bar charts shown in Figure 11.

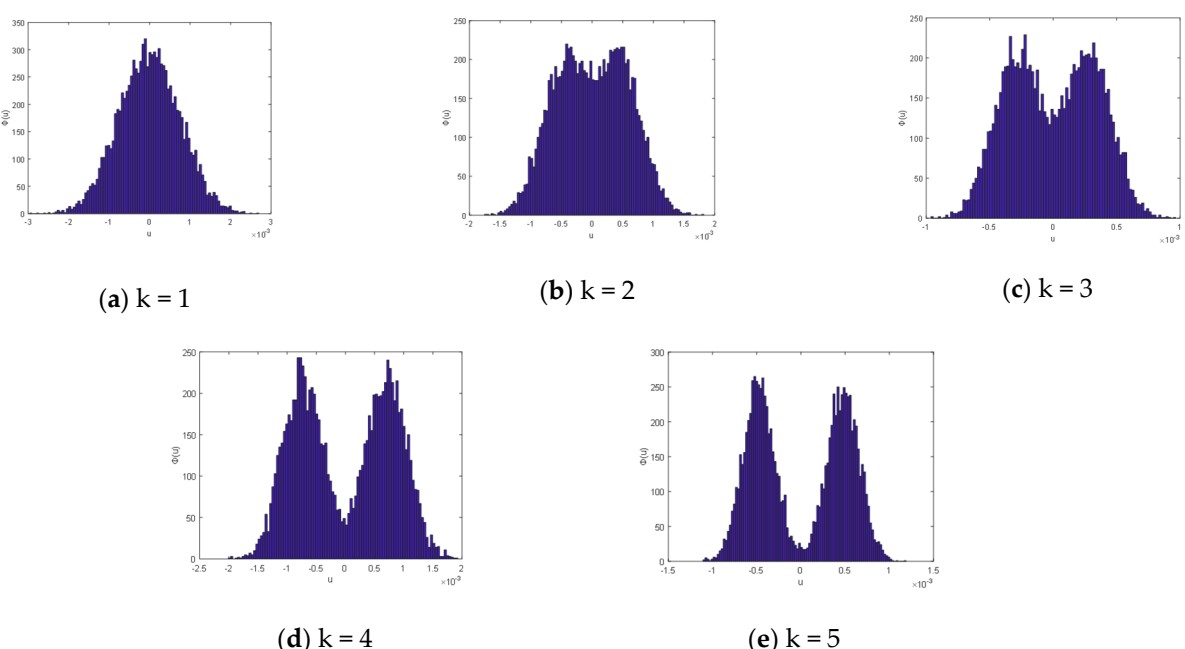

(**a**) k = 1      (**b**) k = 2      (**c**) k = 3

(**d**) k = 4      (**e**) k = 5

**Figure 11.** Average distribution graph of non-Gaussian measurement.

To verify the effectiveness of the proposed generalized M-estimator of optimized parameters, the RMSEs of the estimators obtained from Monte Carlo simulations for different ks and against $\beta$ are shown in Figure 12. The optimized $\beta$, $\beta_{opd}$, can be calculated according to (20) with the obtained A and h. The optimal $\beta$, $\beta_{opl}$, and the RMSEs at $\beta_{opl}$, $\beta_{opd}$, $\beta = 1.5$ for different ks can be obtained from Figure 12. The results are summarized in Table 4.

**Table 4.** Average standard residuals of different estimators in IEEE118 systems at different non-Gaussian noise ratios.

| k | 5 | 4 | 3 | 2 | 1 |
|---|---|---|---|---|---|
| $\beta_{opd}$ | 0.1005 | 0.3804 | 0.6603 | 0.9402 | 1.2201 |
| $\beta_{opl}$ | 0.1001 | 0.1467 | 0.2326 | 0.3687 | 0.9261 |
| RMSE at $\beta_{opd}$ | $1.55 \times 10^{-3}$ | $1.49 \times 10^{-3}$ | $1.45 \times 10^{-3}$ | $1.17 \times 10^{-3}$ | $1.15 \times 10^{-3}$ |
| RMSE at $\beta_{opl}$ | $1.52 \times 10^{-3}$ | $1.43 \times 10^{-3}$ | $1.26 \times 10^{-3}$ | $1.06 \times 10^{-3}$ | $1.11 \times 10^{-3}$ |
| RMSE at $\beta = 1.5$ | $1.82 \times 10^{-3}$ | $1.73 \times 10^{-3}$ | $1.63 \times 10^{-3}$ | $1.32 \times 10^{-3}$ | $1.18 \times 10^{-3}$ |
| ((RMSE at $\beta = 1.5$) $-$ (RMSE at $\beta_{opd}$))/RMSE at $\beta = 1.5 \times 100\%$ | 14.84% | 13.87% | 11.04% | 11.36% | 2.54% |

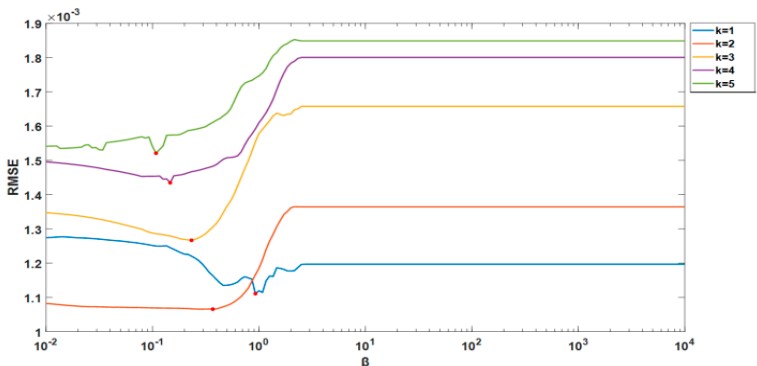

**Figure 12.** The trend of RMSE of each estimator with the value of β in systems with different bimodal interval size systems. The red dot is the nadir.

The analysis performed in Figure 12 and Table 4 leads to the following conclusions:

(1) $\beta_{opd}$ is larger than $\beta_{opl}$, but the RMSE is not much different from the actual RMSE;

(2) The RMSE at $\beta_{opd}$ is significantly lower than the RMSE at $\beta = 1.5$.

The conclusions above demonstrate the effectiveness of the proposed method. The underlying reason for these results is that the proposed method reduces the value of β to increase the estimator's robustness when the level of non-Gaussian measurement noise increases. The linear drooping characteristic of β gives the proposed estimator significantly better estimator accuracy than the traditional generalized M-estimator, but slightly lower estimator accuracy than the estimator with $\beta_{opl}$, which is acceptable considering the difficulty to obtain $\beta_{opl}$.

### 5.3. Simulation Examples of the Polish 2736 Bus System

To verify that which is proposed, simulations are conducted in the Polish 2736 bus test system [27].

Among the total 3269 measurements in the system, 280 of them have bimodal Gaussian distributed noises with the bimodal interval size (k) randomly set between 0~5. According to the proposed optimized parameter selection method, the optimized β, β opd, is determined as follows:

Step 1: Randomly select 800 measurements;

Step 2: For each measurement selected, test the measurement noises and record 10,000 groups of data; 120 non-Gaussian measurement noises are counted according to the statistics, and the proportion of non-Gaussian measurement, *h* is 15%;

Step 3: Calculate the sagging coefficient A;

Step 4: Find the average *k* according to the statistic bar chart of all bimodal Gaussian distributed measurements noises. To verify the effectiveness of the proposed generalized M-estimator of optimized parameters, the RMSEs of the estimators obtained from Monte Carlo simulations for different ks and against β are shown in Table 5.

**Table 5.** Average standard residuals of different estimators in Polish 2736 systems at different non-Gaussian noise ratios.

| k | 5 | 4 | 3 | 2 | 1 |
|---|---|---|---|---|---|
| $\beta_{opd}$ | 0.6751 | 0.8330 | 0.9876 | 1.167 | 1.340 |
| $\beta_{opl}$ | 0.1079 | 0.1467 | 0.2326 | 0.3687 | 0.9261 |
| RMSE at $\beta_{opd}$ | $1.69 \times 10^{-3}$ | $1.54 \times 10^{-3}$ | $1.39 \times 10^{-3}$ | $1.22 \times 10^{-3}$ | $1.10 \times 10^{-3}$ |
| RMSE at $\beta_{opl}$ | $1.61 \times 10^{-3}$ | $1.48 \times 10^{-3}$ | $1.34 \times 10^{-3}$ | $1.16 \times 10^{-3}$ | $1.05 \times 10^{-3}$ |
| RMSE at $\beta = 1.5$ | $2.03 \times 10^{-3}$ | $1.96 \times 10^{-3}$ | $1.78 \times 10^{-3}$ | $1.57 \times 10^{-3}$ | $1.43 \times 10^{-3}$ |
| ((RMSE at $\beta = 1.5$) − (RMSE at $\beta_{opd}$))/RMSE at $\beta = 1.5 \times 100\%$ | 16.74% | 21.43% | 21.91% | 22.29% | 23.08% |

Analysis performed in Table 5 leads to the following conclusions:

(1) The $\beta$ value calculated by sagging coefficient is slightly different from the actual $\beta$ value, but its RMSE is not much different from the actual RMSE;

(2) The RMSE corresponding to the calculated $\beta$ value is significantly higher than that when $\beta = 1.5$, and the RMSE is significantly higher when the *k* value is larger.

These conclusions are almost the same with those in Section 5.2, demonstrating the adaptability of the proposed even in very large networks.

## 6. Conclusions and Future Works

This paper presents a generalized M-estimation method for power system state estimation with sampling-based optimization parameters. The sampling statistics method can set the parameter according to the different non-Gaussian measurement noise parameters and the proportion to adjust its robustness, which increases its overall estimation accuracy. The generalized M-estimation defines standardized residuals based on the median, thus being more robust than ordinary M-estimation methods. Exhaustive simulation results in the IEEE14 bus test system demonstrate the impacts of non-Gaussian measurement noise and bad data on estimation accuracy, which are summarized as follows:

(1) Small $\beta$ can increase the robustness of the estimator but decrease the estimation accuracy in the normal operation case where no bad data occur, and vice versa. There is an optimal $\beta$ that can achieve the highest overall estimation accuracy;

(2) The optimal $\beta$ decreases with the increase of the magnitude of bad data and the non-Gaussian degree of measurement noise;

(3) The selection of the optimal $\beta$ value mainly depends on the non-Gaussian degree and proportion of the measured noise.

Simulations in the IEEE 118 bus test system and Polish 2736 bus system verify that the proposed method can improve the estimation accuracy by up to 23% compared to the traditional generalized M state estimator in the presence of different degrees of non-Gaussian measurement noise.

Although this paper has reached the expected goal, further research can be carried out by considering more newly developed estimators, adopting more complex grid models, and considering more complex situations.

**Author Contributions:** Y.S.: Conceptualization, Methodology, Data curation, Investigation, Software, Writing original draft. Y.H.: Methodology, Validation, Visualization, Writing. Y.Y.: Validation, Visualization, Editing. Z.J.: Supervision, Funding acquisition, Project administration, Writing. M.A.M.: Review & editing. All authors have read and agreed to the published version of the manuscript.

**Funding:** This research was funded by National Natural Science Foundation of China, grant number 51907106.

**Institutional Review Board Statement:** Not applicable.

**Informed Consent Statement:** Informed consent was obtained from all subjects involved in the study.

**Data Availability Statement:** Research data is available at https://drive.google.com/file/d/17QEiRvHyZ38LqEr7hUffad1m9SvLZDn-/view?usp=sharing, https://drive.google.com/file/d/1TDkySmOztwf4i3HJyeMVU16FdUWvm8DZ/view?usp=sharing (accessed on 4 June 2022).

**Conflicts of Interest:** The authors declare no conflict of interest.

## Nomenclature

| | |
|---|---|
| WLS | Weighted least squares |
| LAV | Least absolute value |
| PMU | Phasor measurement unit |
| IRLS | Iterated re-weighted least square |
| BGM | Bimodal Gaussian mixture |
| MLE | Maximum likelihood estimation |
| SCADA | Supervisory control and data acquisition |
| RMSE | Root mean square error |

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
