# Peer review of "Robust Power System State Estimation Method Based on Generalized M-Estimation of Optimized Parameters Based on Sampling"

_sustainability, doi:10.3390/su15032550_

Round 1

Reviewer 1 Report

Several improvements in terms of the English language use need to be made. There is an excessive and redundant use of connectors across the paper. For example, in lines 36 and 37, the connector “However” is used redundantly, which is confusing since two sentences that are supposed to be connected and reinforcing each other end up by contradicting themselves, making harder for the reader to understand the authors’ point of view in the exposed topic. Similar mistakes can be found across the paper.

Line 40, instead of “…may carry or may not carry bad data.” Use a simpler connection such as “… may or may not carry…” or put the comment in terms of the bad data uncertainty. The paragraphs are too long and condensed, please, use tables for presenting your literature review, which can be complemented with a brief introduction to the background the authors are trying to present at the introduction. As presented, is just too hard to follow.

Smooth the connections between paragraphs and sentences, the sentences are not connected and is not possible to follow the topic exposed. It almost feels robotic.

Line 140 -> “reference not found!”. The citations across the document don’t follow the guidelines provided by MDPI, therefore, it is hard to differentiate between a quantity, citation, or equation number across the document.

Author Response

The authors would like to thank the reviewer for the advices to improve this paper. We have carefully checked the manuscript and corrected the usage of "However", the connections between sentences and the mistakes in references. We have also re-formulated the literature review in the form of table. The modified parts are marked in red font.

Reviewer 2 Report

This manuscript presented the Robust Power System State Estimation Method Based on Generalized M Estimation of Optimized Parameter Based on Sampling. Major issues are associated with the manuscript.

1. Abstract, summarize the numerical results of proposed work, and discuss how it outperforms existing works.

2. Related work should be mentioned in a separate section by highlighting the comparative analysis in tabular manner. What are the unique features of this study compared to the existing works?

3. Contributions should be highlighted in bullet points and justified literature

4. A ‘Research Gap’ section should incorporate which will states the purpose of the study.

5. Methodology section is not clear in present form. A flowchart should incorporate which represent the various steps of the proposed work.

6. Figures quality can be improved. 

7. Conclusion also required presenting in more quantitative manner.

8. Follow the journal template. 

9. Authors mentioned that the existing robust estimators mainly consider the suppression performance on bad data, while the outlier suppression performance caused by non-Gaussian measurement noise is insufficient. This results in a decreased robustness of the state estimator in the presence of non-Gaussian measurement noise. How the proposed method remove this disadvantage and at what level.

10. Authors must incorporate the percentage improvement in the robustness by utilizing the proposed estimation.

11. The major issue with he manuscript is missing comparative analysis. A detailed comparative analysis with existing techniques is required which compare the results of proposed method with other techniques.

12. To recommend the proposed work for the practical applications, a real test system must be utilized and proposed method will be implemented on it.

13. A detailed mathematical formulation of the proposed technique must be clearly added into the manuscript.

Author Response

  1. Abstract, summarize the numerical results of proposed work, and discuss how it outperforms existing works.

Response 1: We have revised the Abstract with the summary of numerical results. Thanks for the comment.

  1. Related work should be mentioned in a separate section by highlighting the comparative analysis in tabular manner. What are the unique features of this study compared to the existing works?

Response 2: We have revised the literature review part in the Introduction in the form of table. The unique features of this study compared to the existing work is that it considers the trade-off between robustness and accuracy of the estimator and then proposes a method that can automatically strike a balance between robustness and accuracy.

  1. Contributions should be highlighted in bullet points and justified literature

Response 3: Contributions have been highlighted in bullet points in the Introduction as marked in red font.

  1. A ‘Research Gap’ section should incorporate which will states the purpose of the study.

Response 4: We have added Section 3.3 which presents the purpose of the study.

  1. Methodology section is not clear in present form. A flowchart should incorporate which represent the various steps of the proposed work.

Response 5: We have created flowcharts for the proposed methods, please see Figures 3-4. Thank you for your suggestion.

  1. Figures quality can be improved.

Response 6: We have checked the figures and improved the low quality figures.

  1. Conclusion also required presenting in more quantitative manner.

Response 7: We have revised the conclusion with the results quantitatively analysed.

  1. Follow the journal template.

Response 8: We have downloaded and used the journal template.

  1. Authors mentioned that the existing robust estimators mainly consider the suppression performance on bad data, while the outlier suppression performance caused by non-Gaussian measurement noise is insufficient. This results in a decreased robustness of the state estimator in the presence of non-Gaussian measurement noise. How the proposed method remove this disadvantage and at what level.

Response 9: The proposed method suppresses the errors caused by non-Gaussian measurement noise by appropriately choosing the value of β so that the outliers caused by non-Gaussian measurement noise are given less weights while the normal data can be accurately estimated. This is demonstrated in Figure 6, which shows that a minimum estimation error can be achieved for each non-Gaussian measurement noise at certain value of β.

  1. Authors must incorporate the percentage improvement in the robustness by utilizing the proposed estimation.

Response 10: Robustness is defined as an estimator’s ability to resist abnormal data such as bad data and outliers caused by non-Gaussian measurement noise. An estimator is said to be more robust than the another estimator if it can detect abnormal data that the other estimator fails to detect. However, it is difficult to quantitatively assess the robustness of the estimators since the sensitivity to abnormal data is also decided by the magnitude of the abnormal data and the types of abnormal data. For example, in [R1] the robustness of its proposed robust estimator is assessed by whether or not it can detect and suppress bad data, different types of non-Gaussian measurement noises and strong system nonlinearity rather than quantitatively assessed in percentage. Since for the same type of robust estimator its robustness is determined by its choice of parameters, the robustness of the proposed estimator can be seen from the value of β, and the smaller the value of β, the better robustness the estimator has. As can be seen from Equation (19), the β is mostly smaller than 1.5, showing that the robustness of the proposed estimator is generally higher than the traditional GM estimator based on Huber M estimator.

[R1] J. Zhao and L. Mili, "A Robust Generalized-Maximum Likelihood Unscented Kalman Filter for Power System Dynamic State Estimation," in IEEE Journal of Selected Topics in Signal Processing, vol. 12, no. 4, pp. 578-592, Aug. 2018, doi: 10.1109/JSTSP.2018.2827261.

  1. The major issue with he manuscript is missing comparative analysis. A detailed comparative analysis with existing techniques is required which compare the results of proposed method with other techniques.

Response 11: The manuscript compares the proposed method with the traditional WLS estimator, LAV estimator and the traditional GM estimator based on Huber M estimator. When β=0, infinity and 1.5, the estimator is equivalent to these 3 types of estimators, respectively. The comparative results can be easily seen from the figures and tables in Section 5.

  1. To recommend the proposed work for the practical applications, a real test system must be utilized and proposed method will be implemented on it.

Response 12: We have added one sub-section in Section 5 presenting the simulation results in a real Polish 2736 bus test system.

  1. A detailed mathematical formulation of the proposed technique must be clearly added into the manuscript.

Response 13: We have added a few more equations for the derivation of the method proposed in the flowcharts (Figures 3-4) in the manuscript. Hopefully it will better facilitate the understanding of the proposed method.

Reviewer 3 Report

Comments and Suggestions for Authors

Article entitled with “Robust Power System State Estimation Method Based on 2 Generalized M Estimation of Optimized Parameter Based on 3 Sampling” the authors proposed a power system state estimation method for generalized M-estimation of the optimized parameters based on sampling. The proposed method optimized parameter determination method improves the accuracy of state estimation while ensuring robustness. Although research work looks interesting; however, the following are the shortcoming that must be addressed to enhance the article quality and clarity for the readers.

§  The abstract must be revised and improved. It is lacking of indicating measurable achievements obtained from the proposed model.

§  The Introduction / background section also needs to be improved with latest research work proposed in this research area.

§  Author should mention clearly about the novelty of the paper and contributions in the introductory section and in the abstract.

§  It would be better if all the abbreviations and notations used in the mathematical models are described clearly as a separate sub-section

§  Section 5.1, Line 297, “…..results shown in this chapter……..” This is not a book chapter, need correction

§  Simulation study is a good initiative; however, experimental validation of proposed methodology with real-world application is required.

§   The conclusion section should be revised to link it with the abstract in a better way.

§  The English proofreading and improvement are highly recommended.

Author Response

  •  The abstract must be revised and improved. It is lacking of indicating measurable achievements obtained from the proposed model.

Response 1: We have revised the Abstract with the summary of numerical results. Thanks for the comment.

  •  The Introduction / background section also needs to be improved with latest research work proposed in this research area.

Response 2: A few more relevant latest references have been added into literature review. Thanks for the comment.

  •  Author should mention clearly about the novelty of the paper and contributions in the introductory section and in the abstract.

Response 3: We have re-written the abstract and the introduction with highlights of the contributions of the paper. Thanks for the suggestion to improve this paper.

  •  It would be better if all the abbreviations and notations used in the mathematical models are described clearly as a separate sub-section

Response 4: We have added a nomenclature before the Introduction which explains all abbreviations and notations used in the mathematical models. Thanks for the advice.

  •  Section 5.1, Line 297, “…..results shown in this chapter……..” This is not a book chapter, need correction

Response 5: We have corrected this mistake by changing “chapter” to “section”. Thanks for pointing out this mistake.

  •  Simulation study is a good initiative; however, experimental validation of proposed methodology with real-world application is required.

Response 6: We have added one sub-section in Section 5 presenting the simulation results in a real Polish 2736 bus test system.

  •   The conclusion section should be revised to link it with the abstract in a better way.

Response 7: We have revised the conclusion with better link to the abstract. Thanks for the advice.

  •  The English proofreading and improvement are highly recommended.

Response 8: We have gone through the manuscript carefully and improved the English language, grammar, punctuation, spelling, and overall style. I hope the revised manuscript will meet the requirements of academic publishing in MPDI.

Round 2

Reviewer 1 Report

Thanks for sending your revised paper. I can see several improvements in the paper's structure, which definitely facilitate going through the content and appreciate the contribution claimed by the authors. Still there are some style mistakes that can easily be corrected in the final publication of the article.

For instance, my only observation is on the abstract since, the authors state initially that their algorithm is aimed to improve state estimation robustness, however, later on it is stated that the robustness is adjusted to avoid affecting the estimation's accuracy, which suggests robustness of the algorithm as a function of its accuracy, ergo, the algorithm robustness improvement is conditioned and limited depending on the case. This should be discussed widely and it is my thinking that it should be the paper's core, that if I was able to understand the paper correctly.

Author Response

Response: We would like to give sincere thanks to the reviewer for the recognition of our work and the pungent comment on this paper. There are different methods to improve the robustness of the estimator. Because of their different principles, some of the them affect the estimation accuracy in the normal operation condition, while some do not. The generalized-M estimator is a combination of two methods: projection statistics method and M-estimator. Projection statistics allows the estimator to use the median value of the data instead of the mean value of the data as the center of the data so that the center of the data will not be biased by the outlier and the distance between the outlier and the center of the data can be more accurately quantified. This helps improve the robustness of the estimator without compromising the estimation accuracy. M-estimation down-weights the outliers whose distances are greater than the given threshold. The robustness of M-estimation is decided be the threshold. Reducing the threshold can increase the  robustness the estimator, but decrease the estimation accuracy since the corresponding loss function deviates more from the loss function of maximum likelihood estimation which is quadratic. We have modified relative statements to clarify the confusions, the modified parts are marked in red font.

Reviewer 2 Report

Authors tried to incorporate comments of the reviewer but still some major issues are associated with the manuscript which must be addressed before further processing of the manuscript. 

1. Figures' quality still very poor. Figure 5 and figure 10 must be expanded for the better understanding. Furthermore, size of figure 2, and figure 11, must be increased for understanding of the labels of x and y a=xis.  moreover, there is some formatting issue in figure 12 due to that it is not visible in the manuscript. make it correct. 

2. Table 5 is also not in adequate form. Make it correct. 

3. Still the comparative analysis is not clear form the presented results in the section 5. authors must clearly mention that which table or figures provided the comparative analysis and under which case. Furthermore, authors mention that the comparative analysis is presented with traditional WLS estimator, LAV estimator and the traditional GM estimator, why not authors utilized recent advanced estimators for the comparative analysis purpose. 

Author Response

  1. Figures' quality still very poor. Figure 5 and figure 10 must be expanded for the better understanding. Furthermore, size of figure 2, and figure 11, must be increased for understanding of the labels of x and y a=xis.  moreover, there is some formatting issue in figure 12 due to that it is not visible in the manuscript. make it correct.

Response 1: We have expanded Figure 5, and Figure 10, and significantly increased the size of Figure 2 and Figure 11 so that the axes can be seen clearly. We have also modified Figure 12. Hopefully it can be seen clearly this time. Thanks very much for these detailed comments on the figure quality of the paper.

  1. Table 5 is also not in adequate form. Make it correct.

Response 2: We have revised Table 5, thanks for the comment.

  1. Still the comparative analysis is not clear form the presented results in the section 5. authors must clearly mention that which table or figures provided the comparative analysis and under which case. Furthermore, authors mention that the comparative analysis is presented with traditional WLS estimator, LAV estimator and the traditional GM estimator, why not authors utilized recent advanced estimators for the comparative analysis purpose.

Response 3: We have revised Section 5 and added clear reference to the tables and figures where the comparative analysis is provided. It is true that the many advanced estimators have been proposed recently. We chose to study the performance of GM estimator for the following reasons. 1. There is no point to compare estimators that improve different performances of state estimation.  For example, some data driven state estimators based on machine learning improve the estimation performance by improving the forecasting prediction [R1] or by allowing to estimate the network which is not fully observable [R2]. Even for the particular performance of robustness, different estimators improve it with different principles. For example, some estimators propose methods to detect and remove the outliers according to the characteristics of the estimation results; while some estimators down-weight the outliers according to the loss function. 2. M estimator is a type of robust estimators that have been widely adopted by many state estimators. Thus, studying M estimation and improving its performance can benefit a wide range of state estimators.

[R1] Y. Chen, H. Chen, Y. Jiao, J. Ma and Y. Lin, "Data-driven Robust State Estimation Through Off-line Learning and On-line Matching," in Journal of Modern Power Systems and Clean Energy, vol. 9, no. 4, pp. 897-909, July 2021, doi: 10.35833/MPCE.2020.000835.

[R2] G. Tian, Y. Gu, D. Shi, J. Fu, Z. Yu and Q. Zhou, "Neural-network-based Power System State Estimation with Extended Observability," in Journal of Modern Power Systems and Clean Energy, vol. 9, no. 5, pp. 1043-1053, September 2021, doi: 10.35833/MPCE.2020.000362.

[R3] J. Yang, W. -A. Zhang and F. Guo, "Distributed Kalman-Like Filtering and Bad Data Detection in the Large-Scale Power System," in IEEE Transactions on Industrial Informatics, vol. 18, no. 8, pp. 5096-5104, Aug. 2022, doi: 10.1109/TII.2021.3119136.

[R4] 21.J. Zhao, M. Netto and L. Mili, “A robust iterated extended Kalman filter for power system dynamic state estimation”, IEEE Transactions on Power Systems, vol. 32, no. 4, pp. 3205-3216, 2017.

Reviewer 3 Report

The revised version is significantly improved, and results are validated which enhanced the clarity and efficacy of proposed approach

Author Response

Thank you very much for your recognition on this paper!

Round 3

Reviewer 2 Report

Authors tried to incorporate most of the comments raised by the reviewer. still some minor issues are remaining.

1. Define the labels of x and y axis in the figure 1. 

2.  Define the labels of x and y axis in the figure 11. 

3. Discussion section should be expanded based on the presented results and comparative analysis. 

Author Response

Thanks very much for the additional comments provided by the reviewer to further improve our paper. We have defined the labels in Figures 1 and 11, and also added more discussions in Section 5.